# Byte Pair Encoding for Symbolic Music

**Nathan Fradet**[1,2]**, Nicolas Gutowski**[3]**, Fabien Chhel**[4,3]**, Jean-Pierre Briot**[1]

[1]Sorbonne University, CNRS, LIP6, F-75005 Paris
[2]Aubay, Boulogne-Billancourt, France
[3]University of Angers, LERIA, 49000 Angers, France
[4]ESEO, ERIS, 49100 Angers, France

## Abstract

When used with deep learning, the symbolic music modality is often coupled with language model architectures. To do so, the music needs to be tokenized, i.e. converted into a sequence of discrete tokens. This can be achieved by different approaches, as music can be composed of simultaneous tracks, of simultaneous notes with several attributes. Until now, the proposed tokenizations rely on small vocabularies of tokens describing the note attributes and time events, resulting in fairly long token sequences, and a sub-optimal use of the embedding space of language models. Recent research has put efforts on reducing the overall sequence length by merging embeddings or combining tokens. In this paper, we show that Byte Pair Encoding, a compression technique widely used for natural language, significantly decreases the sequence length while increasing the vocabulary size. By doing so, we leverage the embedding capabilities of such models with more expressive tokens, resulting in both better results and faster inference in generation and classification tasks. The source code is shared on Github[1], along with a companion website[2]. Finally, BPE is directly implemented in MidiTok[3], allowing the reader to easily benefit from this method.

## 1 Introduction

When used with deep learning, the symbolic music modality is mostly represented as discrete and used with language models (LM) such as Transformers (Vaswani et al., 2017). These models receive sequences of tokens as input, convert them to learned embedding vectors representing their semantic features in a continuous space, and process these embeddings for the task at hand. A token is a distinct element, known within a finite vocabulary. For natural language, a token can be a word, subword or punctuation mark. For symbolic music, tokens usually represent note attributes or time events, such as pitch or duration. Tokenizing music, i.e., converting raw data into tokens, can be achieved by several ways, as music can be composed of simultaneous tracks, of simultaneous notes with several attributes such as their pitch and duration. Multiple approaches exist to represent these features.

Recently, the token representation of symbolic music has been studied (Fradet et al., 2023), with the goal to improve 1) the results, e.g. the quality of generated music or the accuracy of classification tasks, and; 2) the efficiency of the models. The former is tackled with more expressive representations (Huang and Yang, 2020; Kermarec et al., 2022; von Rütte et al., 2023; Fradet et al., 2021), and the latter by representations based on either token combinations (Payne, 2019; Donahue et al., 2019), or embedding pooling (Hsiao et al., 2021; Zeng et al., 2021; Ren et al., 2020; Dong et al., 2023), which reduce the overall sequence length.

Still, these tokenizations are based on tokens only representing the values of time events and note attributes. This comes with a big limitation: these tokens do not carry much information by themselves. We can assume that their embeddings does not either. By analogy to natural language, these tokens are closer to the character level than word level. Yet, a powerful feature of LMs is their ability to learn (embedding) representations of discrete elements such as tokens, and leverage this information for reasoning and downstream tasks. For natural language, LMs are usually coupled with vocabularies containing up to 50k tokens, represented on a few hundreds dimensions (often between 512 and 2048). Using a vocabulary containing fewer tokens than the number of dimensions used to represent them appears as a suboptimal usage of such models. Moreover, the expressive information carried by music is deduced from the combinations of its notes and their attributes. Considering the infinite

---

[1]https://github.com/Natooz/bpe-symbolic-music
[2]https://Natooz.github.io/BPE-Symbolic-Music/
[3]https://github.com/Natooz/MidiTok

possible music arrangements, we can suppose that using solely note attribute embeddings imposes to LMs a heavier modeling effort than embeddings of potential whole note successions that would be more expressive and explicit.

In this paper, we show that Byte Pair Encoding (BPE, described in Section 3) applied to symbolic music allows to address the two goals just mentioned, while outperforming the previous methods and making the model learn better distributed embeddings. To the best of our knowledge, BPE has yet not been studied for the symbolic music modality, although it can be applied on top of any music tokenization that does not perform embedding pooling. This work aims at closing this gap by shedding light on the results and performance gains of using BPE:

- We experiment on four public datasets (Wang et al., 2020b; Kong et al., 2021; Ens and Pasquier, 2021; Hung et al., 2021), with two base tokenizations, on which BPE is learned with several vocabulary sizes, on generation and classification tasks;

- We compare BPE with other sequence reduction techniques introduced in recent research;

- We study the geometry of the learned embeddings, and show that BPE can improve their isotropy and space occupation;

- We show some limits of BPE, such as on the proportion of sampled tokens, and that the vocabulary size has to be carefully chosen.

## 2  Related work

### 2.1  Tokenization of symbolic music

Most deep learning models using symbolic music generation use a specific music tokenization. Early research introduced representations specifically tied to the training data being used, such as DeepBach (Hadjeres et al., 2017), FolkRNN (Sturm et al., 2015) or BachBot (Liang et al., 2017). Non-autoregressive models such as MuseGAN (Dong et al., 2018) often represent music as pianoroll matrices.

Since, more universal representations have been studied, allowing to convert any sequence of (simultaneous) notes into tokens (Oore et al., 2018; Huang and Yang, 2020; Hadjeres and Crestel, 2021; Fradet et al., 2021). Some of them are depicted in Figure 1.

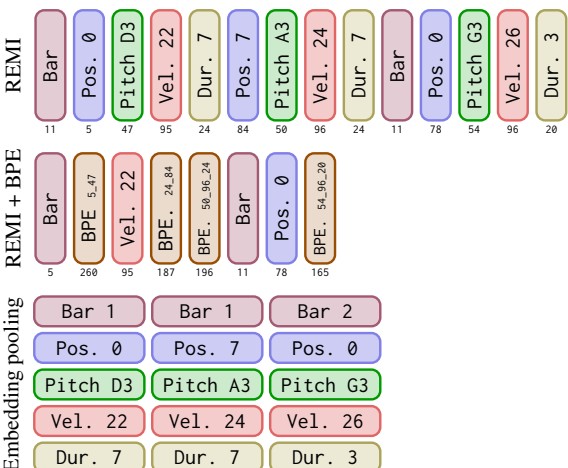

Figure 1: Three tokenizations of the same three notes. Tokens are ordered from left to right, the numbers put below are their integer ids. Top row is *REMI* (Huang and Yang, 2020), middle correspond to the top row with BPE applied to some tokens, bottom row is a tokenization similar to (Zeng et al., 2021; Dong et al., 2023) where the embeddings are merged (pooled).

### 2.2  Sequence length reduction strategies

More recent works put efforts on the efficiency of the models. As most of them rely on the Transformer architecture (Vaswani et al., 2017) and the attention mechanism, their time and space complexity grows quadratically with the input sequence length. This bottleneck led researchers to work on more efficient attention estimations (Tay et al., 2021), down to linear complexity. In the field of symbolic music specifically, researchers worked on strategies to reduce the sequence length in order to increase 1) the efficiency of the models; 2) the scope of the attention mechanism; 3) the quality of the generated results. These strategies can be split in two categories:

- **embedding pooling** such as *Compound Word* (Hsiao et al., 2021) (*CPWord*), *Octuple* (Zeng et al., 2021), PopMag (Ren et al., 2020), SymphonyNet (Liu et al., 2022) or MMT (Dong et al., 2023). Embeddings of several tokens are merged with a pooling operation. This is often done by concatenating the embeddings and projecting the vector, resulting in an aggregated embedding of fixed size.

- **token combination** such as in MuseNet (Payne, 2019), LakhNES (Donahue et al., 2019) or other recent works (Liu et al., 2022; Thickstun et al., 2023), which consists of using a vocabulary of tokens representing sev-

eral attributes, e.g., `Pitch-x_Dur-y` representing both the pitch and velocity information. BPE can be seen as a learned token combination technique.

## 2.3 Limitations

One of the main limitation of the previous work is the suboptimal usage of the embedding space of LMs. Most of them use models with embeddings represented from 512 to 1024 dimensions, for vocabularies of less than 500 tokens. As the model contextually learns to gather embeddings along dimensions representing learned features, using a number of dimensions larger than the number of elements to represent causes embeddings to not take advantage all the space of the embedding dimensions, which will stay unoccupied for a large proportion. For comparison, the same models, when trained on natural language data, use to learn up to 50k embeddings on 512 to 1024 dimensions.

The sequence length reduction strategies just introduced also have big limitations. Embedding pooling: 1) requires specific model input and output modules, which can break compatibility with popular software libraries; 2) requires multiple losses at training, which increases the complexity; 3) for generation, inferring from such model can be seen as sampling from a multivariate distribution, which can be very delicate, as 4) the results can easily degenerate if the pooling does not yield semantically rich embeddings that represent the underlying tokens. On the other hand, token combinations of entire types of tokens can lead to large vocabularies with unused tokens and potentially non-optimized or unbalanced token distributions.

To the best of our knowledge, no work has been conducted on applying BPE (introduced in Section 3) to symbolic music generation. Although (Liu et al., 2022) introduced a method which they named MusicBPE, this technique links weakly with BPE and has a limited scope. It adds to the vocabulary new tokens for recurrent chords. These tokens represent pitch combinations for simultaneous notes having the exact same velocity and duration. It can only be used for a limited proportion of notes (and in turn tokens), actually less than a quarter when a strong downsampling is applied (Appendix D). As it does not apply on token successions, it cannot capture the contextual and probability relations between them, including time dependencies. For these reasons, we do not compare it with BPE as it would not be relevant.

## 3 Byte Pair Encoding

Byte Pair Encoding (BPE) (Gage, 1994) is a data compression technique. It converts the most recurrent successive bytes in a corpus into newly created ones. For instance, in the character sequence aabaabaacaa, the sub-sequence aa occurs three times and is the most recurrent one. Learning and applying BPE on this sequence would replace aa with a new symbol, e.g., d, resulting in a compressed sequence dbdbdcd. The latter can be reduced again by replacing the db subsequence, giving eedcd. In practice BPE is learned on a corpus until the vocabulary reaches a target size. BPE learning is described by the pseudo-code of Algorithm 1.

---

**Algorithm 1** Learning of BPE pseudo-code

---

**Require:** Base vocabulary $\mathcal{V}$, target vocabulary size $N$, dataset $\mathcal{X}$
1: **while** $|\mathcal{V}| < N$ **do**
2:     Find $m = \{t_1, t_2\}$, the most recurrent token succession from $\mathcal{X}$
3:     $\mathcal{V} \leftarrow \mathcal{V} + [t_{|\mathcal{V}|} : m]$
4:     Substitute occurrences of $m$ in $\mathcal{X}$ with $t_{|\mathcal{V}|}$
5: **end while**
6: **return** $\mathcal{V}$

---

BPE is nowadays largely used in the NLP field as it allows to encode rare words and segmenting unknown or composed words as sequences of sub-word units (Sennrich et al., 2016). Other token aggregation, or vocabulary building techniques exist. The two other most commonly used are Unigram (Kudo, 2018) or WordPiece (Wu et al., 2016), which operations share similarities with BPE.

For natural language, bytes are the distinct characters composing the text. For symbolic music, the distinct note and time attributes can be used as the "bytes" to merge. In this context, BPE can allow to represent a note, or even a succession of notes, that is recurrent in the dataset, as a single token. For instance, a note that would be tokenized as the succession of tokens `Pitch_D3`, `Velocity_60`, `Duration_2.0` could be replaced by a single new one. Rare note (and attributes) can still be tokenized as non-BPE tokens. The same logic applies to time tokens, that can also be associated to note tokens.

## 4 Experimental settings

### 4.1 Models and training

As we specifically focus on LMs, we experiment with the state of the art deep learning architecture for most NLP tasks at the time of writing, the Transformer (Vaswani et al., 2017). We use the GPT2 (Radford et al., 2019) and BERT (Devlin et al., 2019) implementations of the transformers library (Wolf et al., 2020) for respectively music generation and classification. They are made of 12 layers, embedding sizes of 512, eight attention heads and feed-forward layers of 2048. They count approximately 40M learned parameters. The generators are trained with teacher forcing with the target sequence being the input shifted by one to the left. The classifier are first pretrained to retrieve randomized tokens, and then finetuned to classify the input sequences. More details on the training procedure can be found in Appendix A.

All models receive sequences of 256 to 384 tokens, beginning with special BOS (Beginning of Sequence) and ending EOS (End of Sequence) tokens. We split datasets in three subsets: one only used for training to update the models, one for validation during training, one used to test the models after training. The last two represent respectively 10% and 15% of the dataset for classification and 2% and 5% for generation.

### 4.2 Tokenization

We experiment with *REMI* (Huang and Yang, 2020) and *TSD* (for Time Shift Duration) as base tokenizations, on top of which BPE is applied. Both tokenizations describe notes as a succession of Pitch, Velocity and Duration tokens. *REMI* represents time with Bar and Position tokens, which respectively indicates when a new bar is beginning and at which position within the time is. *TSD* represents time with TimeShift tokens, indicating explicit time movements. For the multitrack MMD dataset, we prepend a Program token before the Pitch token of each note to represent its instrument.

When tokenizing symbolic music, continuous characteristics are usually downsampled to discrete sets of values (Huang and Yang, 2020; Oore et al., 2018; Hadjeres and Crestel, 2021). For instance, velocities can be downsampled from 128 to 32 values. These sets should be sufficiently precise to keep the global information. Downsampling these characteristics helps models to learn more easily, as the values of the reduced sets will be more

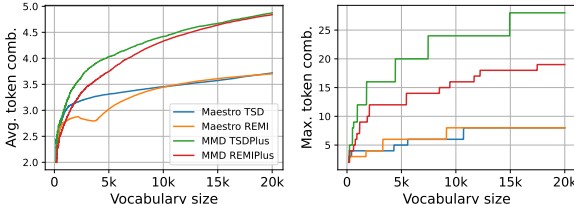

(a) Avg. nb. of combinations  (b) Max. nb. of combinations

Figure 2: Average and maximum number of token combinations of tokens learned with BPE in function of the vocabulary size.

| Strategy | Voc. size | | tokens/beat (↓) | | Tok. time (↓) | | Detok. time (↓) | |
|---|---|---|---|---|---|---|---|---|
| | TSD | REMI | TSD | REMI | TSD | REMI | TSD | REMI |
| No BPE | 149 | 162 | 18.5 | 19.1 | 0.174 | 0.151 | 0.031 | 0.039 |
| BPE 1k | 1k | 1k | 9.3 (-49.5%) | 10.4 (-45.3%) | 0.187 | 0.163 | 0.053 | 0.063 |
| BPE 5k | 5k | 5k | 7.0 (-62.2%) | 8.5 (-55.2%) | 0.181 | 0.165 | 0.053 | 0.064 |
| BPE 10k | 10k | 10k | 6.3 (-66.0%) | 7.7 (-59.7%) | 0.183 | 0.164 | 0.052 | 0.065 |
| BPE 20k | 20k | 20k | 5.8 (-68.9%) | 6.9 (-63.9%) | 0.184 | 0.163 | 0.052 | 0.063 |
| PVm | 1453 | 1466 | 13.4 (-27.8%) | 13.8 (-27.4%) | 0.134 | 0.123 | 0.024 | 0.026 |
| PVDm | 28185 | 28198 | 8.2 (-55.5%) | 8.6 (-54.8%) | 0.119 | 0.106 | 0.025 | 0.030 |
| CP Word | | 188 | | 8.6 (-54.8%) | | 0.169 | | 0.034 |
| Octuple | | 241 | | 5.2 (-72.6%) | | 0.118 | | 0.035 |

Table 1: Vocabulary size, average tokens per beat ratios, and average tokenization and decoding times in second using MidiTok (Fradet et al., 2021) and the Hugging Face tokenizers[4] libraries, on the Maestro dataset.

distinctive. We detail our downsampling strategy in Appendix C.

We choose to experiment with five vocabulary sizes: without BPE, 1k, 5k, 10k and 20k tokens.

Finally, we compare BPE with other sequence reduction strategies introduced in Section 2.2. We experiment with merging Pitch and Velocity tokens (*PVm*), and Pitch, Velocity and Duration together (*PVDm*). *PVm* is similar to the strategy used with MuseNet (Payne, 2019). We also experiment with embedding pooling strategies - *CPWord* (Hsiao et al., 2021) and *Octuple* (Zeng et al., 2021) - that we group with *REMI* in our experiments as they represent time similarly. We use the same pooling strategy, and sample independently from the logits of each output modules. All embeddings have the same size than the model dimension.

## 5 BPE learning

BPE allows to significantly reduce the sequence length. As shown in Figure 2, the ratio of average number tokens representing a beat can be reduced up to more than 50%. As BPE replaces recurrent pair of bytes in the data, the average number of byte combinations of the vocabulary tends to first quickly increase, then more slowly grow. The inverse tendency can be observed on the tokens per

---

[4] https://github.com/huggingface/tokenizers

beat ratios shown in Table 1, while showing that BPE increase only slightly the tokenization time. The maximum number of byte combinations varies depending on the data. Here, the MMD dataset allows to learn much more combined tokens. This shows that the Maestro dataset contain much diverse token successions, which is not surprising considering that it is made of classical music while MMD contains many genres, among which some with very repetitive patterns. The tokenization time with BPE naturally increases, but stays relatively close to the baselines without.

Appendix E complements this analysis by shedding light on the types of the underlying tokens represented by the newly learned tokens.

# 6 Music generation

Music generation is a popular application of deep learning models (Briot et al., 2020; Briot, 2021). We ought to experiment on this task to demonstrate the benefits of BPE on music modeling. For this task, we choose to use the Maestro dataset (Hawthorne et al., 2019), which is made of 1k pairs of audio and MIDI files of classical piano performances. Each MIDI file is made of one piano track, with dynamic melodies and complex harmonies. We generate autoregressively the next 512 tokens of input prompts from the test subsets of the GiantMIDI dataset, filtering the logits by keeping the top $p = 0,95$ probability mass (nucleus sampling (Holtzman et al., 2020)) and top 15 token probabilities (top-k sampling (Fan et al., 2018)).

Evaluation of symbolic music is still an open issue (Yang and Lerch, 2020). In the absence of automatic metrics measuring the distances between subsets of data, most works evaluate generated results with human surveys along with feature similarity metrics. The latter however cannot capture the quality of music, and is subject to irregularities in case of model over or underfitting. We decide here to replace them with an metric measuring the errors of prediction of the models.

## 6.1 Tokenization syntax error

Every music tokenization has an underlying syntax of token type and value successions, that can normally happen. For instance, if the last token of an input sequence is of type `Pitch`, some tokenization could imply that the next token to be predicted must be of type `Velocity`. We could also expect a model to not predict more than once the same note

at a same time, or to not go back in time.

Successions of incorrect token types can be interpreted as errors of prediction. These errors can help us to measure if a model has efficiently learned the music representation and if it can yield coherent results, or not otherwise. From this motivation, we introduce a new metric we called Tokenization Syntax Errors (TSE).

We distinguish five categories of errors:

- **TSE$_{type}$**: the predicted token is of an invalid type regarding the previous one;

- **TSE$_{time}$**: a predicted `Position` value is inferior or equal to the current one, making the time goes backward;

- **TSE$_{dupn}$** (duplicated note): when the model predicts a note that has already been played at the current moment (by the same instrument);

- **TSE$_{nnof}$** (no NoteOff): when using `NoteOn` and `NoteOff`, and that a `NoteOn` token has been predicted with no `NoteOff` later to end it, or too distant in time;

- **TSE$_{nnon}$** (no NoteOn): when a `NoteOff` token is predicted but the corresponding note has not been played.

For a given sequence of tokens, TSE measures the ratio, scaled between 0 and 1, of errors for these five categories. A TSE of 0 means that there is no error in the sequence, while a ratio of 1 means only errors were predicted. Our experiments are not concerned by the last two categories as we do not use `NoteOff` tokens.

Finally, we should mention that most of these errors can be avoided by a ruled-based sampling. When predicting a token, it is possible to track the time, notes played and token types to automatically exclude invalid predictions. In practice, this can be achieved by setting the invalid indices of the predicted logits to $-\infty$ before softmax.

## 6.2 Human evaluations

For both *TSD* and *REMI* tokenizations, we selected about 130 prompts of 4 bars from the test subset, and generated continuations of 512 tokens with all models. We gathered nine participants, among which seven are musicians, to evaluate the results. They were asked to open the MIDI files with Digital Audio Workstation (DAW) softwares such as Logic Pro or Ableton, play each track individually

| | TSE$_{type}$(↓) | | TSE$_{dupn}$(↓) | | TSE$_{time}$(↓) | | Hum. Fidelity (↑) | | Hum. Correctness (↑) | | Hum. Diversity (↑) | | Hum. Overall (↑) | |
|---|---|---|---|---|---|---|---|---|---|---|---|---|---|---|
| **Strategy** | TSD | REMI | TSD | REMI | TSD | REMI | TSD | REMI | TSD | REMI | TSD | REMI | TSD | REMI |
| **No BPE** | 1.53 | 1.34 | 4.19 | 5.59 | - | **28.93** | 4.9% | 4.0% | 2.0% | 2.0% | 1.0% | 0.0% | 4.8% | 0.0% |
| **BPE 1k** | 1.59 | 0.62 | 3.60 | 4.16 | - | 34.65 | 13.6% | 11.9% | 11.8% | 14.9% | 10.8% | 6.8% | 8.6% | 8.6% |
| **BPE 5k** | **0.31** | **0.38** | 3.28 | 4.10 | - | 39.25 | 21.4% | **31.7%** | 20.6% | 21.8% | 11.8% | 11.7% | 20.0% | 18.1% |
| **BPE 10k** | 0.49 | 1.04 | 3.83 | 6.39 | - | 48.16 | 23.3% | 20.8% | **29.4%** | 22.8% | 18.6% | 20.4% | 22.9% | 29.5% |
| **BPE 20k** | 0.38 | 0.64 | 4.09 | **3.60** | - | 52.00 | **29.1%** | 19.8% | **29.4%** | 24.8% | **36.3%** | **34.0%** | **30.5%** | **30.5%** |
| **PVm** | 2.45 | 2.99 | 16.90 | 16.33 | - | 36.31 | 2.9% | 2.0% | 2.9% | 0.0% | 7.8% | 2.9% | 4.8% | 1.0% |
| **PVDm** | 0.63 | 6.32 | **2.84** | 10.64 | - | 46.75 | 4.9% | 9.9% | 3.9% | 11.9% | 13.7% | 21.4% | 8.6% | 12.4% |
| **CPWord** | | 6.15 | | 28.55 | | 62.15 | | 0.0% | | 2.0% | | 2.9% | | 0.0% |
| **Octuple** | | - | | 244.11 | | 305.43 | | 0.0% | | 0.0% | | 0.0% | | 0.0% |

Table 2: Metrics of generated results. TSE results are all scaled at $e^{-3}$ for better readability. Hum stand for human, "-" for non-concerned (i.e. 0).

| | tok/sec (↑) | | beat/sec (↑) | | note/sec (↑) | | Voc. sampled (↑) | |
|---|---|---|---|---|---|---|---|---|
| **Strategy** | TSD | REMI | TSD | REMI | TSD | REMI | TSD | REMI |
| **No BPE** | 40.2 | 43.8 | 4.5 | 9.9 | 10.6 | 10.9 | 100% | 100% |
| **BPE 1k** | 78.5 | 67.0 | **13.0** | 17.9 | 20.8 | 16.8 | 100% | 99.9% |
| **BPE 5k** | 99.1 | 83.9 | 12.8 | **30.0** | 26.7 | 20.7 | 100% | 99.8% |
| **BPE 10k** | 97.5 | 85.4 | 12.5 | 26.0 | 26.3 | 21.3 | 99.9% | 99.9% |
| **BPE 20k** | **115.6** | **91.7** | 12.9 | 24.9 | **31.5** | 22.7 | 99.4% | 99.7% |
| **PVm** | 59.3 | 58.1 | 8.2 | 12.2 | 15.9 | 14.9 | 99.3% | 99.0% |
| **PVDm** | 89.7 | 87.3 | 11.4 | 17.1 | 24.7 | 23.4 | 75.9% | 74.3% |
| **CPWord** | | 75.8 | | 15.2 | | 19.0 | | 76.7% |
| **Octuple** | | - | | 14.3 | | **58.5** | | 57.4% |

Table 3: Inference speeds on a V100 GPU and a batch size of 64 and ratio of vocabulary sampled during generation. For tok/sec, the results account for "basic" tokens of note attributes and time. Tok/sec for Octuple is not calculated as the equivalent number of base tokens is not clearly deducible.

and select the best one on four criteria: 1) fidelity on pitch scale and rhythm regarding the prompt; 2) correctness, i.e. featuring good note succession and harmony; 3) coherent diversity, i.e. featuring diverse correct melodies and harmonies; 4) their overall subjective preference. The advantage of using DAW software is twofold: it allows to conveniently listen the different tracks, and compare them by also visualizing them as pianorolls. You can find more details on the human evaluations in Appendix F, and all the generated samples used on the demo website (Section 1).

### 6.3 Results and analysis

The TSE and human preferences results are reported in Table 2.

As BPE creates new tokens that combine several types and values, it also increases the overall complexity of music modeling when using these tokens. Thus, we initially expected the generative models to predict higher ratios of errors. But surprisingly, it decreases these ratios, except for the time errors with *REMI*. These results show that the models easily capture the information of the tokens, and that the vocabulary can be scaled consequently.

We gathered total of 814 human preferences, with a bit more than 100 for each criteria for *TSD* and *REMI*. There is a clear preference for results with BPE, especially with vocabularies of 10k and 20k tokens. Baselines without BPE still accounts for a few preferences for the fidelity and correctness criteria, but are less preferred overall, especially with *REMI*. We note that the *PVDm* baselines show competitive preferences with BPE baselines, especially for diversity. Octuple and CP Word perform poorly on the other hand, which is not surprising as they are not 100% autoregressive, and the sense of the combinations of tokens sampled unconditionally is likely to degenerate, especially when time and notes are handled all at once. Overall, BPE helps models to generate more natural and pleasant music. The new contextually learned embeddings may represent richer and more explicit information, helping to model the musical information.

Besides results quality, the second big advantage of BPE is the inference speed increase. We reported three inference metrics - tokens, beat and note per second - in Table 3, along with the proportion of the vocabulary ever sampled by the models.

We first highlight that models with BPE, up to the maximum vocabulary size tested here, do use most of the tokens of the vocabulary, with a slight decrease as the vocabulary grows. This also supports that the vocabulary can easily be scaled while keeping tokens that are still used by the models.

BPE increases all inference speeds measured by at least two times, even with small vocabularies. We note that the increase of beat/sec does not increase linearly with the vocabulary size, which also indicates that the models predict a higher number of notes per beat. CP Word, despite having low tokens per beat ratios (Table 1), yields lower tokens per second generation speeds, due to the additional input and several sampling steps.

| | Genre (↑) | | Artist (↑) | |
|---|---|---|---|---|
| **Strategy** | TSD | REMI | TSD | REMI |
| **No BPE** | 0.836 | 0.796 | 0.907 | 0.876 |
| **BPE 1k** | 0.882 | 0.871 | 0.934 | 0.920 |
| **BPE 5k** | 0.901 | 0.875 | 0.933 | 0.925 |
| **BPE 10k** | **0.904** | 0.869 | **0.937** | 0.922 |
| **BPE 20k** | 0.851 | 0.877 | 0.909 | 0.923 |
| **PVm** | 0.853 | 0.810 | 0.905 | 0.886 |
| **PVDm** | 0.875 | 0.818 | 0.914 | 0.893 |
| **Octuple** | - | **0.923** | - | **0.941** |

Table 4: Average accuracy of classification models.

# 7 Classification

For our classification task, we experiment with the MMD dataset (Ens and Pasquier, 2021). It is, to our knowledge, the biggest MIDI dataset publicly available. It features more than 430k MIDI files of all genres of music with multiple tracks. Each piece is matched to Spotify and MusicBrainz ids, allowing to link them with a wide variety of information such as artist or music genre. In order to get a more quality training corpus, we perform a preprocessing step which deduplicates the files of the same music and keeps only the best. This is explained in Appendix B. We also merged the tracks of the instruments of the same class in order to reduce the overall complexity (Appendix C).

To handle multiple tracks, we placed `Program` tokens before each `Pitch` token of each note to specify its instrument. This strategy is similar to REMIPlus (von Rütte et al., 2023).

We perform genre and artist classification, from the 40 and 100 most present genres and artist in the MMD dataset. The results, reported in Table 4, show that BPE improves the models performances compared to the baselines without, and outperform the other token combination techniques. The models seem to benefit from larger vocabulary sizes. It however shows limits, as the accuracy does not increase from a vocabulary of 10k to 20k tokens. The Octuple baseline outperforms the others. Here, the model is bidirectional (no attention mask) and we do not sample from multiple distributions. Our assumption is that the reduced sequence length allows to carry more information within a same number of tokens, allowing the models to capture more easily the global melody, harmony and music structure and directly improving their performances.

It concurs with our results, and is explored in the next section by analyzing the geometry of the learned embeddings.

| | Isoscore (↑) | | | | PCA ID (↑) | | | | FisherS ID (↑) | | | |
|---|---|---|---|---|---|---|---|---|---|---|---|---|
| | **Gen / Maestro** | | **Pt. / MMD** | | **Gen / Maestro** | | **Pt. / MMD** | | **Gen / Maestro** | | **Pt. / MMD** | |
| **Strategy** | TSD | REMI | TSD | REMI | TSD | REMI | TSD | REMI | TSD | REMI | TSD | REMI |
| **No BPE** | 0.899 | 0.883 | 0.925 | 0.730 | 62 | 66 | 44 | 45 | 5.4 | 5.2 | 8.1 | 7.9 |
| **BPE 1k** | 0.919 | 0.953 | 0.981 | 0.986 | 100 | 99 | 113 | 102 | 7.3 | 6.7 | 15.5 | 12.2 |
| **BPE 5k** | 0.965 | 0.962 | 0.989 | 0.989 | 131 | 119 | 145 | 119 | 9.0 | 8.6 | 16.7 | 13.7 |
| **BPE 10k** | 0.973 | 0.973 | 0.991 | 0.993 | 132 | 118 | 164 | 118 | 9.8 | 9.6 | 18.3 | 15.2 |
| **BPE 20k** | 0.976 | 0.981 | **0.993** | **0.995** | 146 | 122 | 187 | 137 | 10.8 | 10.5 | 21.4 | 16.9 |
| **PVm** | **0.987** | **0.989** | 0.961 | 0.961 | 71 | 67 | 52 | 52 | 7.1 | 6.8 | 13.9 | 14.7 |
| **PVDm** | 0.945 | 0.942 | 0.898 | 0.909 | 38 | 39 | 98 | 87 | 4.4 | 4.4 | **24.1** | **22.8** |

Table 5: Isoscore, and intrinsic dimension (ID) estimations. Gen. corresponds to the causal generative models, Pt. to the pretrained bidirectional models.

# 8 Learned embedding spaces

We have shown so far that BPE improves the results of music modeling on the generation and classification tasks. Our assumption is that, non-only the reduced sequence length allows to pack more information (longer music piece) within the same number of tokens, but mostly the vocabulary can be scaled while making the model efficiently learn and use the embedding representations of the newly created tokens with BPE.

The embedding space, i.e. the way LMs learn to represent tokens into a continuous space $\mathbb{R}^d$ of $d$ dimensions, has recently been studied (Gao et al., 2019; Biś et al., 2021; Cai et al., 2021). More specifically, it has been shown that most LMs learn anisotropic embedding distributions (Ethayarajh, 2019; Reif et al., 2019), despite that their isotropy have been linked to improved performances on downstream tasks (Gong et al., 2018; Wang et al., 2020a; Biś et al., 2021; Liang et al., 2021; Rajaee and Pilehvar, 2022).

Isotropy is a measure of the uniformity of the space occupied by a manifold across all dimensions. A high isotropy is associated with a uniform variance of the distances between the points of the manifold across all dimensions. In our case the manifold is the collection of contextually learned embeddings $X \in \mathbb{R}^{V \times d}$ where $V$ is the vocabulary size and $d$ the model/embedding dimension. An isotropic embedding space can be viewed as a space where the embeddings are uniformly spaced with uniform densities.

Isotropy is often estimated by different ways: singular value decomposition (SVD) (Biś et al., 2021; Gao et al., 2019; Liang et al., 2021; Wang et al., 2020a), intrinsic dimension (Cai et al., 2021), partition function (Arora et al., 2016; Mu and Viswanath, 2018), average cosine similarity (Ethayarajh, 2019). We chose the two firsts, along with IsoScore (Rudman et al., 2022) which alleviates some of their shortcomings, to have results

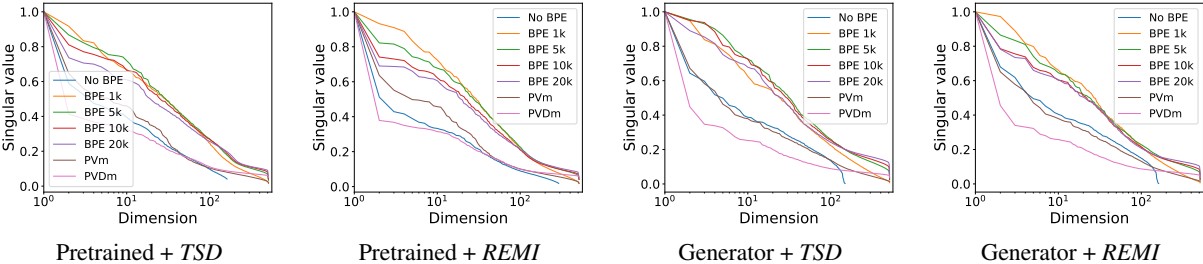

Figure 3: Normalized singular values of the embedding matrices. Pretrained refers to the (bidirectional) classification models after pretraining, and generators to the (causal) models for generation after training.

that complement themselves. We did not measure isotropy on models using embedding pooling, as it would be untractable considering the very large number of possible embeddings, and that the low frequency of the majority of them would result in unreliable results.

SVD, for which results are plotted in Figure 3, allows to visualize the relative domination of some dimensions. Baselines without BPE and *PVm* and *PVDm* show quicker singular value decays, indicating that fewer dominate, whereas baselines with BPE show more uniformly distributed values.

The intrinsic dimension is an estimation of the minimal number of dimensions $n$ required to represent a manifold in $\mathbb{R}^d, d > n$. It links with isotropy in the sense that an isotropic manifold occupies all the dimensions, hence its intrinsic dimension is close to $d$. We chose to estimate it with the Principle Component Analysis (PCA) (Fukunaga and Olsen, 1971) and FisherS (Albergante et al., 2019) methods as they are insensitive to redundancy, focus on common similarities and can scale to large number of points and dimensions. As the embedding matrix is often initialized with a stochastic method around the origin, a simple method can estimate high intrinsic dimensions even though the points coordinates have been very little or not even optimized. This can be the case when a large number of tokens has low frequencies or are absent from the training data, such as with *PVDm*. The intrinsic dimensions results are reported in Section 8, along with the IsoScores. In all cases, as the vocabulary grows with BPE, the intrinsic dimension increases, the embeddings occupy more space.

IsoScore is an estimation of isotropy based on the distance of the covariance matrix of a Principle Component Analysis (PCA) and the identity matrix, and is normalized between 0 and 1. As for the intrinsic dimension, the isoscore grows with the vocabulary size, indicating that the embeddings are

more uniformly distributed.

We also note that similarly to models trained on natural language (Ethayarajh, 2019), our bidirectional models learn more isotropic embeddings than causal (generative) ones. Appendix G depicts UMAP representations of the embedding, showing the narrow cones and clusters they form.

## 9 Conclusion

We showed that BPE can increase the quality of results of Transformer models for symbolic music generation, and classification tasks, while significantly improving their efficiency and inference speed and making better use of their embedding spaces. BPE can be applied on top of any tokenization. The tokenization and decoding times are almost not affected by this extra step, when performed by a well-optimized Rust code. Considering the considerable benefits and low requirements of this technique, we advise anyone using Transformer models with symbolic music to use BPE.

There are still questions that remain uncovered. We showed that 40M parameters models can handle well vocabularies up to 20k tokens with medium-size datasets. We however do not know what are the limits in vocabulary and dataset sizes over which the results might not improve. Moreover, we experimented with BPE, but other common vocabulary building techniques exist, such as Unigram (Kudo, 2018) and WordPiece (Wu et al., 2016). Recent work on natural language showed that Unigram yielded higher model performances than BPE (Bostrom and Durrett, 2020), it might also be the case for symbolic music. Future research will study these questions and hopefully find optimal tokenization guidelines to improve model performances under more various settings.

## Limitations

BPE allows to build vocabulary based on data. Hence, in case the data has specific token distributions, a model trained with this vocabulary might not generalize and perform well on data with opposite token distributions.

BPE implies an additional step during data tokenization. In Table 1 we showed that the impact on tokenization time is very limited. The impact on decoding time is however more substantial.

Finally, although we experimented with two public datasets, two tokenizations and two tasks, we did not find a "limit" vocabulary size above which the results might not increase with. More research must be performed in order to find such limit.

## Ethics Statement

We believe that open science and open sourcing code and model parameters ensure an equal access to the latest research to everybody. Nevertheless, we acknowledge that generative models can be used in harmful ways to artists and copyright owners. Generative models can be used to create new content, that can be conditioned on human prompt such as text description. Malevolent users might control them to copy, alter or use content of artist without their approval. Moreover, such model can represent an unfair competitive tool to music creators, which is a time of writing an open issue and subject to ethic considerations.

## Acknowledgements

This work was partially funded by Aubay. We would like to thank Eric Remilleret for is helpful support.

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

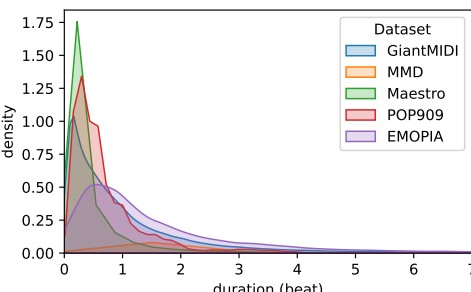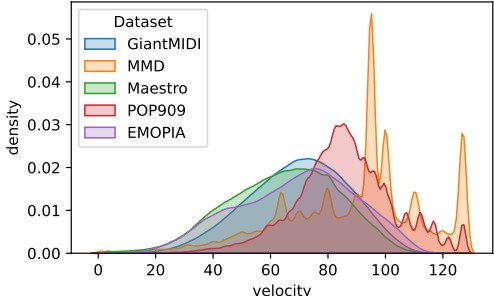

Figure 4: Distributions of the note durations and velocities of five popular MIDI datasets. The duration axis is limited to 7 beats for better readability.

## A  Model training

The generator and classifiers are respectively trained and pretrained on 100k steps. For classifiers pretraining, we use the same objective than done with BERT (Devlin et al., 2019): 15% of each input sequences is masked with a special MASK token, 10% of these masked tokens are randomized, and the loss is computed on the capacity of the model to recover the original tokens. Additionally each sequence is divided into two equal parts separated with a special SEP token, and half of the batch sequences have non-related parts. The model predicts if the second part is the next part of the first. The input embedding and output pretraining module weights are tied to improve the performances (Press and Wolf, 2017).

The classifiers are then finetuned on 10k steps on the downstream tasks. We feed the output hidden state of the first position (BOS token) to an output fully connected layer, to train the model to classify the input sequence.

Trainings are performed on V100 and RTX2080ti GPUs, each time in distributed setups of pairs of the same GPU model, for a total batch size of 128. All trainings are done with automatic mixed-precision (Micikevicius et al., 2018), the Adam optimizer (Kingma and Ba, 2015) with $\beta_1 = 0.9$, $\beta_2 = 0.999$ and $\epsilon = 10^{-8}$, and dropout, weight decay and a gradient clip norm of respectively $10^{-1}$, $10^{-2}$ and 3. We use a one cycle learning rate scheduler: the initial learning rate is close to 0 and gradually grows for the 30% first steps to $1e - 4$ for the generators and classifier pretraining and $3e - 5$ for the classifier fine-tuning, then slowly decreases down to 0. The model parameters are saved when the validation loss is the lowest ever observed, and after training the last version saved is used for testing.

## B  MMD preprocessing

With more than 436k MIDI files, the MMD dataset contains many duplicated songs, corrupted files and poor quality performances. In order to train our models with a well balanced dataset composed of pieces of good quality, we perform a preprocessing step to deduplicate each song, and keep the best versions.

Each MIDI file has a matching score with audio files linked to Spotify and MusicBrainz ids. Hence, each MIDI file can have high matching scores with several different ids, and an Spotify or MusicBrainz id can have have high matching scores with several different MIDI files.

In order to deduplicate the songs, we represented the matching scores as a weighted bipartite graph, and computed its matching. To build the graph, we first read each MIDI file, add it to the graph if it is not corrupted, has a $\frac{4}{*}$ time signature and has at least three tracks. The opposite nodes are the Spotify ids, and the edges (weights) are the MIDI-audio matching scores. When the graph is complete, we compute its matching in order to have the maximum sum of the weights between pairs of distinct and unique MIDIs and Spotify ids. After matching, we end up with 30k distinct MIDI files.

## C  Data downsampling

Figure 4 shows the distributions of velocity and duration values of the notes from the two datasets we use, along with the POP909 (Wang et al., 2020b), Maestro (Hawthorne et al., 2019) and EMOPIA (Hung et al.,

|  | POP909 | Maestro | GiantMIDI | MMD | EMOPIA |
|---|---|---|---|---|---|
| Ticks | 0.014 | 0.000 | 0.002 | 0.143 | 0.002 |
| Preprocessed (32nd) | 0.124 | 0.129 | 0.182 | 0.203 | 0.124 |
| Preprocessed (16th) | 0.175 | 0.229 | 0.236 | 0.222 | 0.145 |

Table 6: Ratio of notes played simultaneously with the same velocity. Preprocessed rows means that the onset and offset times in ticks of the notes have been aligned, to the corresponding portion of bar. For a fair comparison, results for POP909 are for all tracks being merged, and those for MMD are for the unprocessed (vanilla) dataset.

2021) datasets which are commonly used in the music information retrieval community. As there is a larger proportion of low note durations (below two beats), we decided to downsample the `Duration` and `TimeShift` tokens with different resolutions: those up to one beat are downsampled to 8 samples per beat (spb), those from one to two beats to 4 spb, those from two to four beats to 2 spb, and those from four to eight beats to 1 spb. This way, short notes are represented more precisely than longer ones, reducing the vocabulary size. For *REMI*, `Position` tokens are downsampled to 8 spb, resulting in 32 different tokens as we only consider the $\frac{4}{*}$ time signature. This allows to represent the $16^{th}$ note. We only consider pitches within the recommended range for piano (program 0) specified in the General MIDI 2 specifications[5]: 21 to 108. We then deduplicate all duplicated notes. Velocities are downsampled to 8 distinct values. No additional token (e.g., *Chord*, *Tempo*) is used.

We finally apply a downsampling on the instruments for the MMD dataset. The General MIDI 2 protocol features 128 instruments, called programs. In practice, many programs are very similar in their sounds and the way they are played. A model could struggle to capture the differences between these similar programs, especially considering that the program choices were made by humans and potentially subject to bias or subjective preferences. In order to reduce alleviate this complexity, we merge the tracks with programs of the same category, for the twelfth first categories (programs from 0 to 95) except ensembles (programs 48 to 55), and drums, ending up with twelve distinct programs.

## D  Proportion of simultaneous notes in common datasets

Table 6 shows the ratios of notes being played simultaneously (having the same onset and offset times), with the same velocity, for the datasets used in this paper, as well as POP909 (Wang et al., 2020b), GiantMIDI (Kong et al., 2021) and EMOPIA (Hung et al., 2021).

The proportion of simultaneous note is low, even with a strong downsampling of their attributes, onset and offset times. Hence, the scope of token aggregation techniques such as in SymphonyNet (Liu et al., 2022) is arguably limited.

---

[5]Available on the MIDI Manufacturers Association website

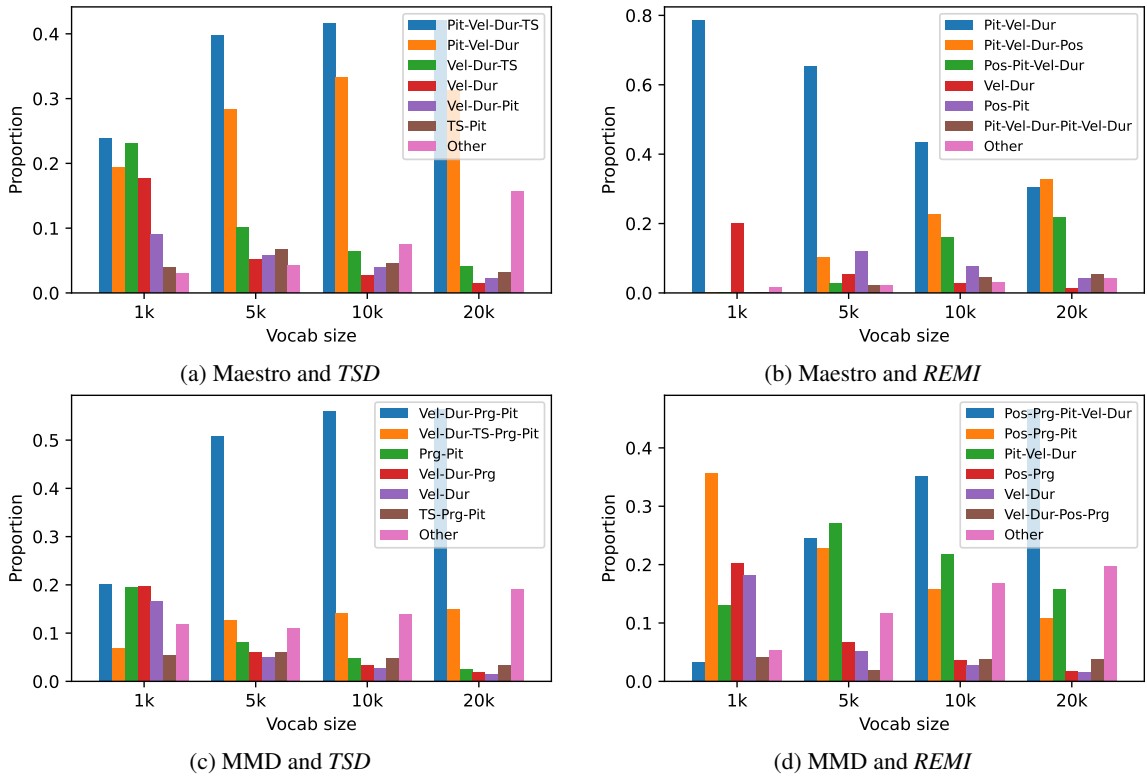

| (a) Maestro and *TSD* | (b) Maestro and *REMI* |
| (c) MMD and *TSD* | (d) MMD and *REMI* |

Figure 5: Normalized distributions of the token types of the BPE tokens, per vocabulary size. Abbreviations in the legend stand for: Pit: Pitch; Vel: Velocity; Dur: Duration; Pos: Position; TS: TimeShift; Prg: Program.

## E   Types of learned byte pairs

Figure 5 shows the distribution of token types combinations of the learned BPE tokens. The majority of the learned combinations represent single notes in all cases, except for the case of MMD when tokenized with *TSD*. In this latter case, most BPE tokens begin with *Velocity* base tokens, indicating that the dataset contains a lot of recurrent *Velocity - Duration* token successions. With *REMI* however, the *Position* token seems to be more recurrent, showing that the notes have more common onset positions, which is not surprising considering that the MMD dataset features many music of genre known to have repeating patterns. As the vocabulary grows, the combinations tend to be more diverse.

## F   Human evaluations

We report here the human evaluation instructions given to the participants to assess the generative models:

*Each MIDI file contains several music tracks generated from different Deep Learning models, that are the continuations of the same 4-bars prompt. For each file, you have to choose the best track on several criteria:*

- *Fidelity: the track with the best fidelity (coherent) relative to the prompt, from a tonal and rhythm point of view;*
- *Correctness: the track with the most correct note successions and harmonies, contrarily to tracks with dissonant notes or unnatural melodies;*
- *Diversity: the track with the best coherent diversity, i.e. featuring diverse correct melodies, contrarily to a music that would repeat the same note patterns. A "bad" or uncertain music (i.e. non-correct) cannot be consider as diverse;*
- *Overall preference: the track that you overall prefer subjectively;*

*Do not answer to all for all the files, as the evaluations can be time-consuming. Fix yourself a number of files to evaluate, and randomly pick them from the list.*

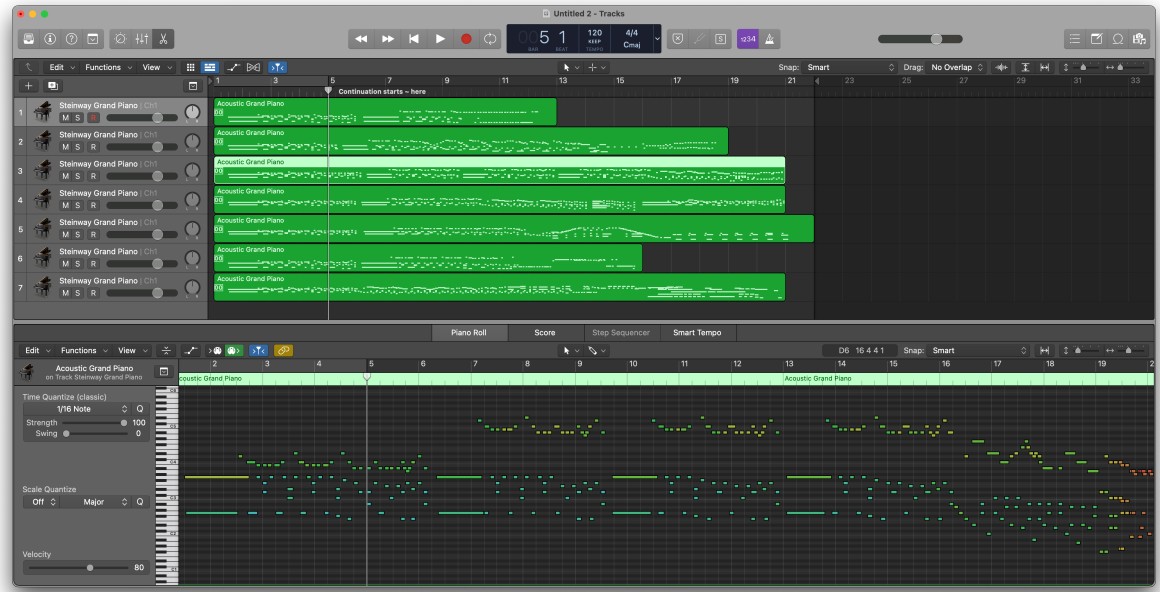

Figure 6: Example of MIDI file given to participants for human evaluations, opened with the Logic Pro DAW.

*You will find generated results than can be very similar, even identical sometimes. As such, you might feel uncertain or unable to decide. In such cases, do not answer for all criteria and just skip to the next file. There is no good or wrong answer, you just have to answer subjectively. Trust yourself and trust your musical instinct.*

An example of MIDI file open with the Logic Pro DAW is shown in Figure 6.

## G  Learned embedding space

UMAP (McInnes et al., 2018) representations shown in Figure 7, Figure 8, Figure 9 and Figure 10 show the embeddings of the models of the paper, computed with the official UMAP Python package with default parameters. For each figure, only 1k randomly sampled points are represented in order to keep them in vector format without adding too much weight in this file document. We encourage the reader to visualize them on our demo website for a more convenient readability.

The models learn clusters of embeddings of the same type. The embeddings do not occupy the space uniformly, but rather have preferred directions following their type and value. We still note that bi-directional (pretrained) models tends to occupy more space than causal (generative) ones. This especially noticeable for the *PVm* and *PVDm* models. For generative models, the embeddings tends to be oriented towards common dimensions, and slightly spread towards orthogonal one.

Pretrained bi-directional models learn more isotropic embedding representations. The embeddings are spread more uniformly across all directions.

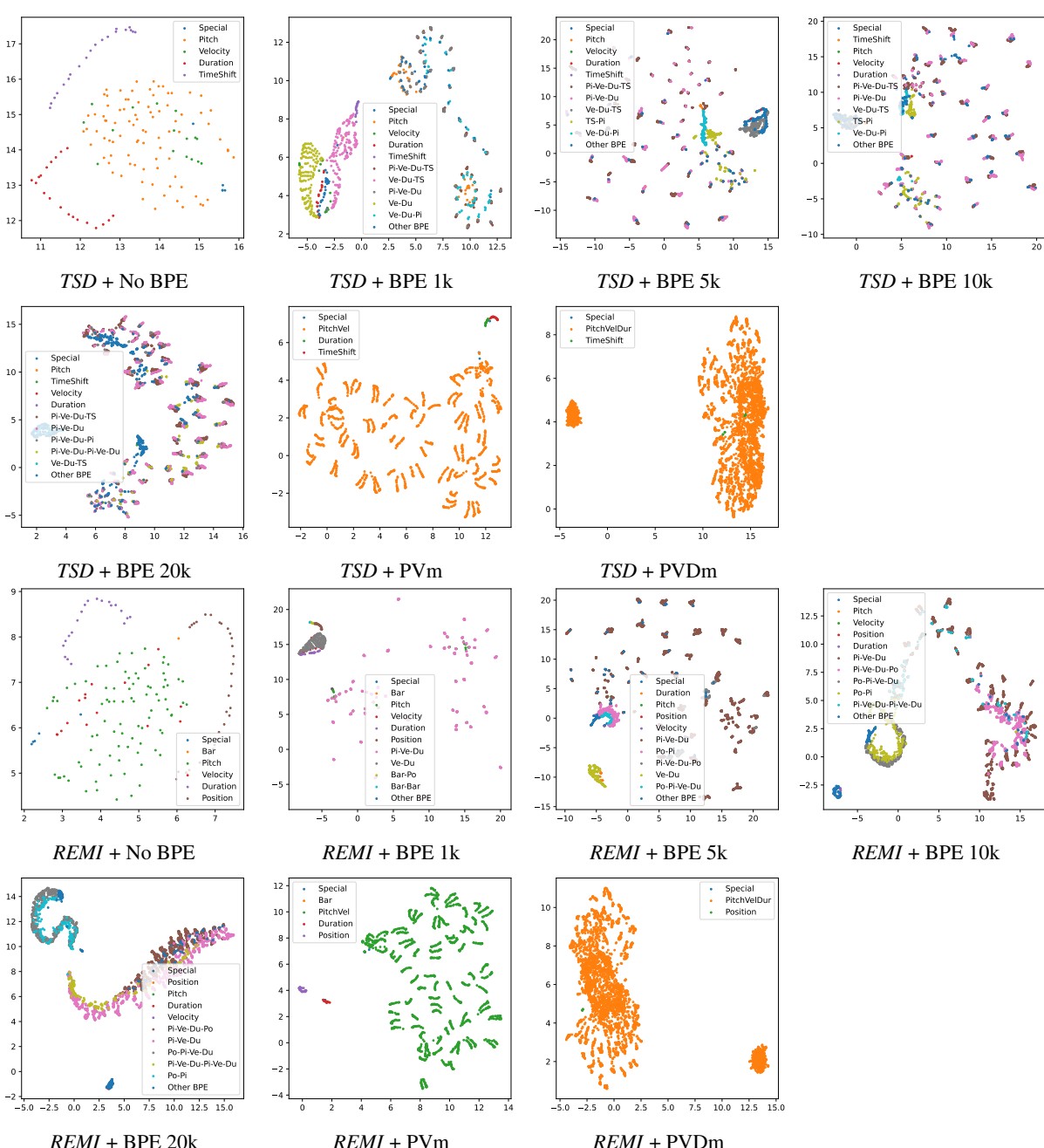

Figure 7: UMAP 2d representations of the embeddings of the generators, trained with the Maestro dataset. Abbreviations in legend stand for: Pi: Pitch; Ve: Velocity; Du: Duration; Po: Position; TS: TimeShift.

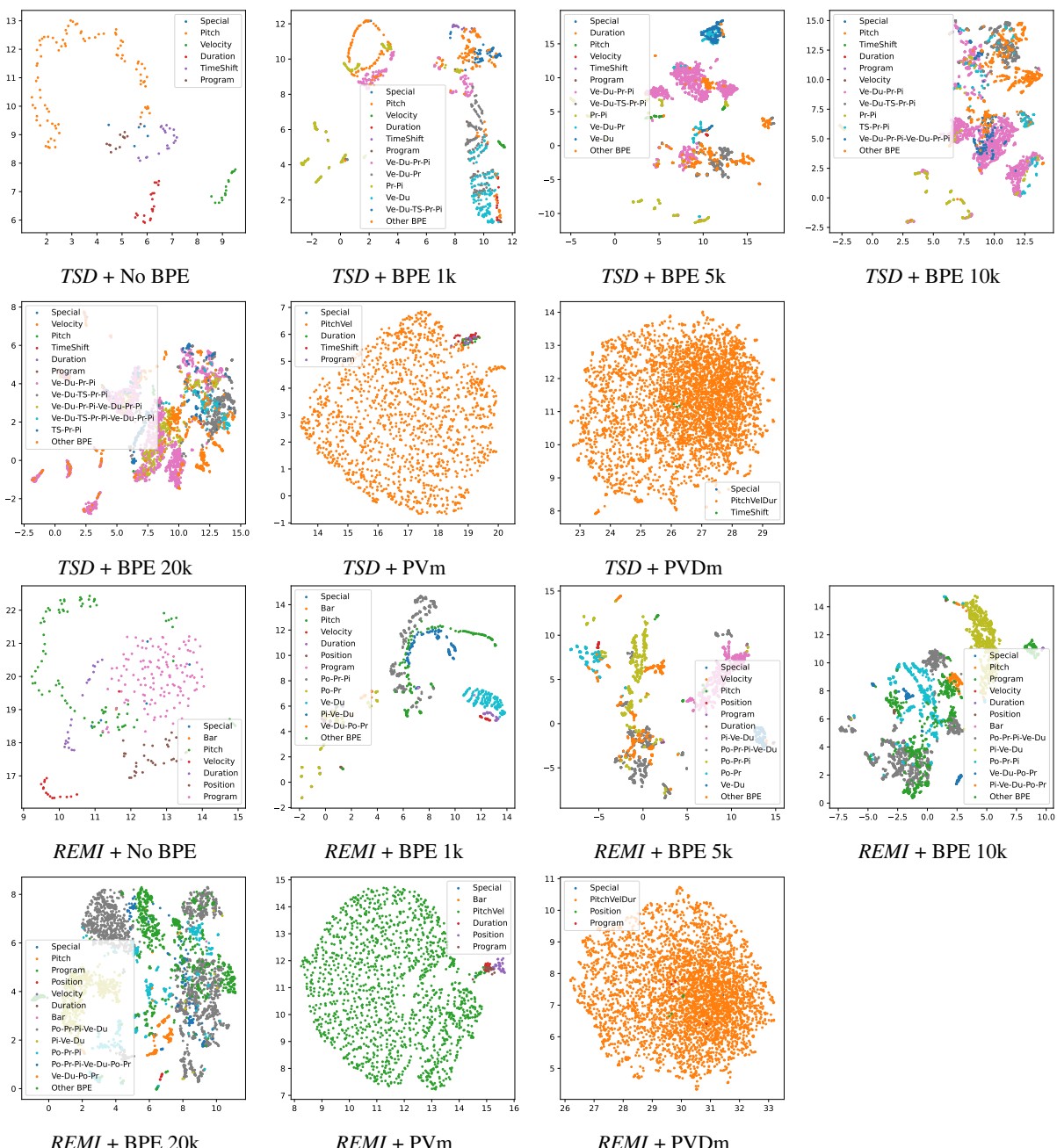

Figure 8: UMAP 2d representations of the embeddings of the pretrained bidirectional models, trained with the Maestro dataset. Abbreviations in legend stand for: Pit: Pitch; Ve: Velocity; Du: Duration; Po: Position; TS: TimeShift; Pr: Program.

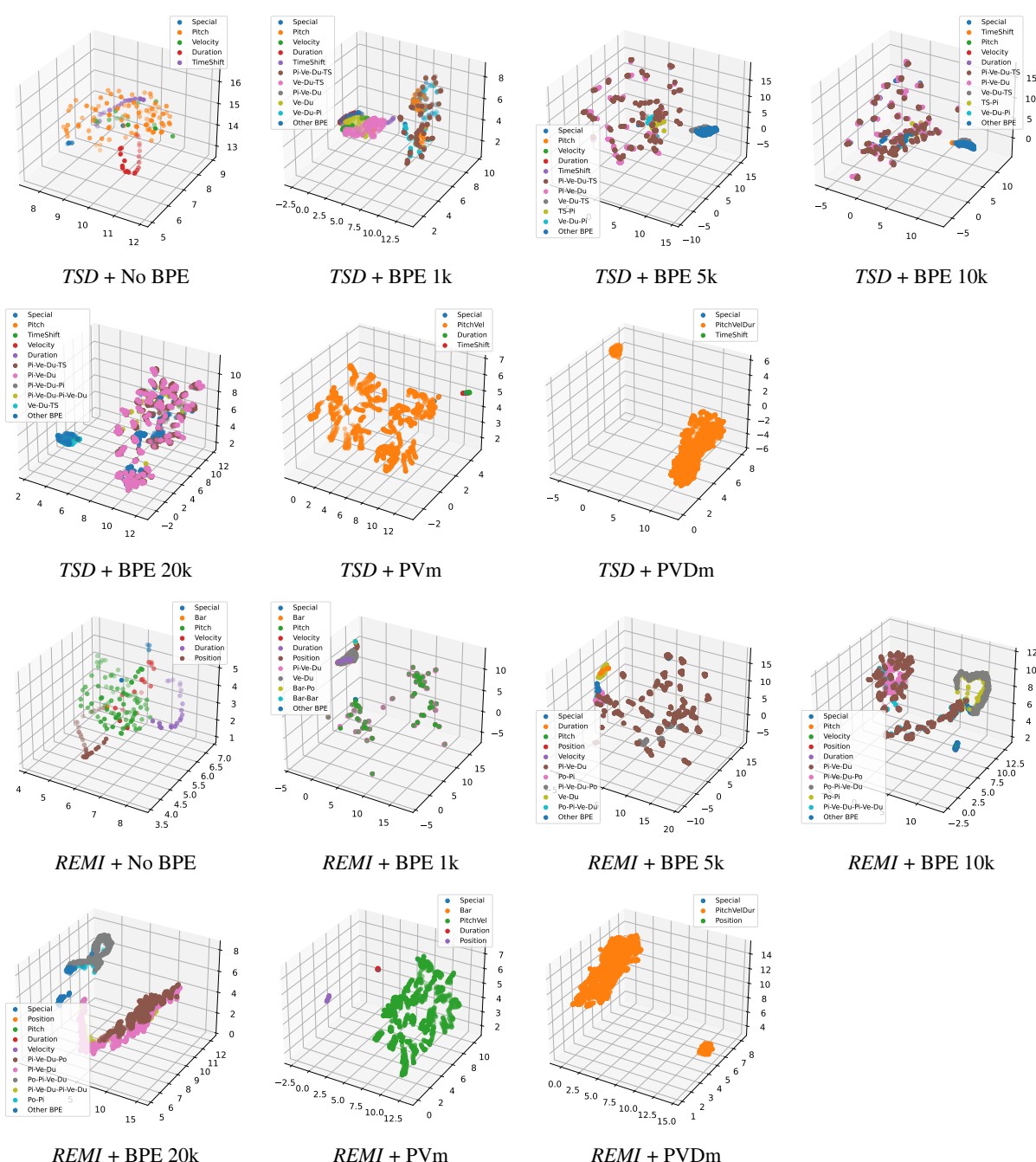

Figure 9: UMAP 3d representations of the embeddings of generative models, trained with the Maestro dataset. Abbreviations in legend stand for: Pi: Pitch; Ve: Velocity; Du: Duration; Po: Position; TS: TimeShift.

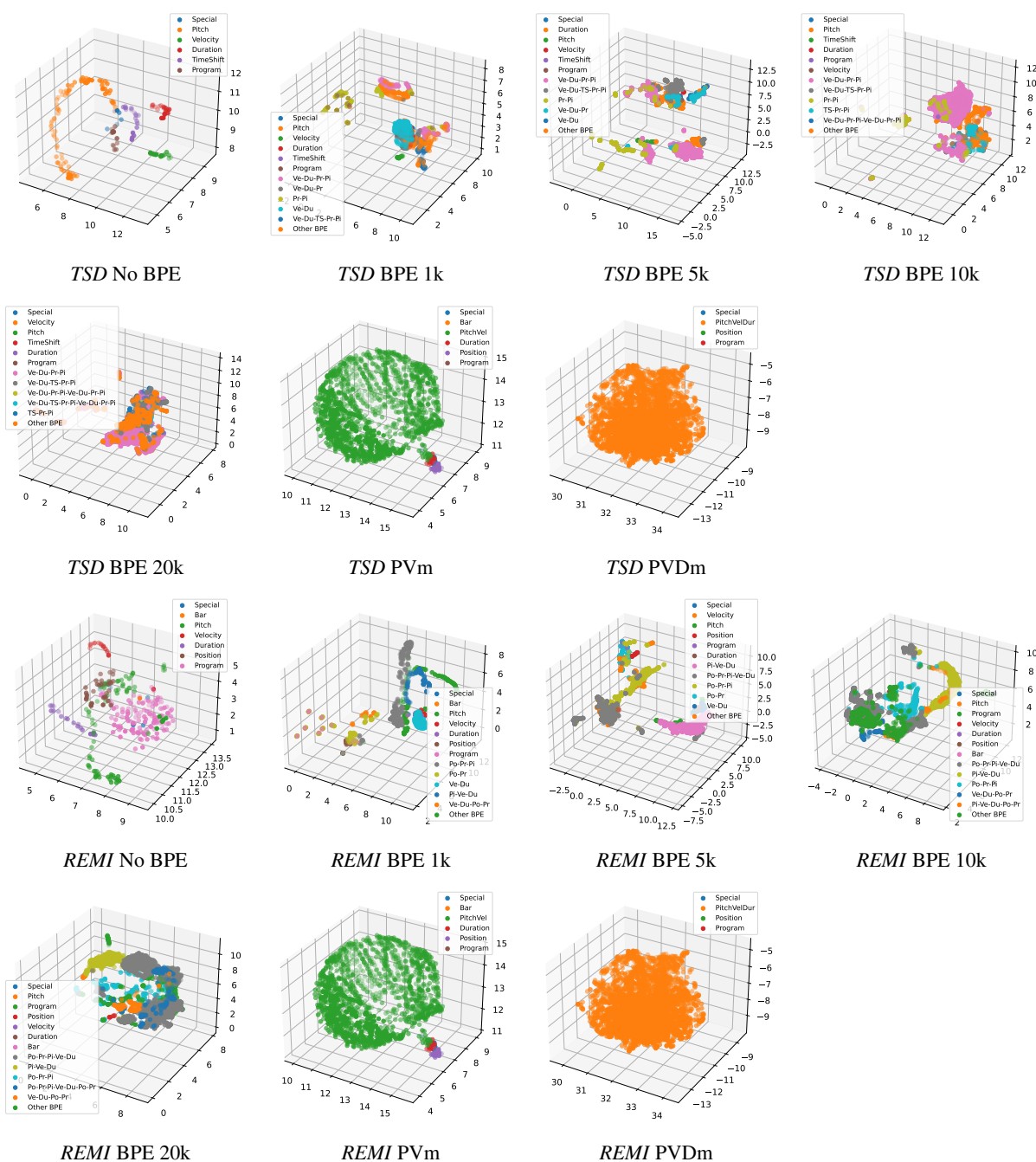

Figure 10: UMAP 3d representations of the embeddings of pretrained bidirectional models, trained with the MMD dataset. Abbreviations in legend stand for: Pi: Pitch; Ve: Velocity; Du: Duration; Po: Position; TS: TimeShift; Pr: Program.