# OpenReview forum: "Byte Pair Encoding for Symbolic Music"
_EMNLP/2023/Conference — EMNLP 2023 Main_

### Official Review · Reviewer_ErZ1 · 2023-07-29

**Soundness:** 4

**Excitement:**

3: Ambivalent: It has merits (e.g., it reports state-of-the-art results, the idea is nice), but there are key weaknesses (e.g., it describes incremental work), and it can significantly benefit from another round of revision. However, I won't object to accepting it if my co-reviewers champion it.

**Paper Topic And Main Contributions:**

The topic of this paper is symbolic music representation.
Authors propose byte pair encoding into symbolic music.

**Reasons To Accept:**

The authors introduce byte pair encoding into symbolic music.
The authors compare BPE with other sequence reduction techniques introduced in recent research.
The authors study the geometry of the learned embeddings, and show that BPE can improve their isotropy and space occupation;
The authors show some limits of BPE.

**Reasons To Reject:**

This paper hardly makes any modifications to the BPE algorithm for music scenarios. Therefore, the novelty of this paper is limited.

**Reproducibility:**

4: Could mostly reproduce the results, but there may be some variation because of sample variance or minor variations in their interpretation of the protocol or method.

**Reviewer Confidence:**

3: Pretty sure, but there's a chance I missed something. Although I have a good feel for this area in general, I did not carefully check the paper's details, e.g., the math, experimental design, or novelty.

---

> ### Author Rebuttal · Authors · 2023-08-23
>
> We agree that our paper does not make substantial modifications to the BPE algorithm, we still emphasize that our contribution lies in the application of this established technique to the realm of music scenarios. By doing so, we show it allows significant performance (inference speed) and results quality gains, at the cost of short preprocessing step. Considering these benefits, we strongly encourage researchers and engineers to use BPE with symbolic music, and so believe that our paper adds valuable insights to the field. Moreover, we make the use of BPE for symbolic music easy for everyone by releasing the code in a friendly and easy way to use, which would encourage researchers to adopt it.
>
> We would be happy to receive your feedback on how we could improve the paper, and in turn convince you to increase your score.

---

### Official Review · Reviewer_ntvA · 2023-08-04

**Soundness:** 4

**Excitement:**

4: Strong: This paper deepens the understanding of some phenomenon or lowers the barriers to an existing research direction.

**Paper Topic And Main Contributions:**

The authors propose to use BPE tokenizer to train a transformer causal to generate and classify symbolic music. The method is said to outperform other tokenization methods on both classification and generation tasks.

**Reasons To Accept:**

The paper is well written, the methodology is solid. There are enough experiments and the method shows to be simple yet effective. The authors provide a nice website with the code and audio samples. The results in generation are impressively better than other methods

**Reasons To Reject:**

The claim on classification performances need to be clearer. The BPE method is not better nor less constraining than Octuple. Especially this sentence needs to be reformulated:
"We advise anyone using Transformer models with symbolic music to use BPE." This would not be true if the goal is to perform classification.

**Reproducibility:**

4: Could mostly reproduce the results, but there may be some variation because of sample variance or minor variations in their interpretation of the protocol or method.

**Reviewer Confidence:**

4: Quite sure. I tried to check the important points carefully. It's unlikely, though conceivable, that I missed something that should affect my ratings.

**Typos Grammar Style And Presentation Improvements:**

In general you should use less adjectives and be more rigorous. Please reformulate the following:

line 072: seems like a big shortfall

line 167: The same models, when used with natural language, use to learn up to 50k embeddings on a range of 512 to 1024 dimensions.

line 165: One can easily see here...

---

> ### Author Rebuttal · Authors · 2023-08-23
>
> Thank you for your recommendations, we will reformulate the typos and formulations.
> We found better classification results with Octuple, however is Octuple is actually more constraining. The reasons are:
>
> * It requires multiple input heads, hence to either modify model implementations (e.g. from Hugging Face transformers, or even the PyTorch Transformer implementation) or create a model from scratch. If you are using a library to train the model, or to generate from it, you might need to modify it too as the input / output tensors have now an additional dimensions that needs to be handled `(batch_size, sequence_length, sub_tokens)`. During pretraining, you also have multiple output heads, the model implementation must be modified too and weight tying [1] must be adapted. This was our case using the HF transformers trainer and generate methods, and we had to spend time to make the adaptations required;
> * Pretraining takes more time (as using multiple losses, one per output module). We did not emphasize on this downside in the paper, as our experiment is quite short (about 10 hours to pretrain), but we can report the training times in the paper;
> * It can break the traceability of models (torchscript);
> * Octuple cannot represent additional tokens such as chords, sustain pedal, pitch bend or control change effects. In theory, one could adapt the tokenizer to represent these features by adding more input/output heads, but this would also increase the complexity of the model and training time, and good results are not guaranteed. Its expressivity is hence limited.
>
> These are the reasons for our recommendation to use BPE, as it allows competitive results, out of the box, on top of any tokenization and without having to deal with these limitations / inconveniences. But we must admit that these reasons were not clearly stated in the paper, that the sentence is not clearly supported, and we thank you for having pointed out this. We plan to add them in the paper, but we are also open to simply reformulate the sentence.
>
> [1] Press, O., & Wolf, L. (2017). [Using the Output Embedding to Improve Language Models.](https://aclanthology.org/E17-2025/) In Proceedings of the 15th Conference of the European Chapter of the Association for Computational Linguistics: Volume 2, Short Papers (pp. 157–163). Association for Computational Linguistics.

---

### Official Review · Reviewer_heEW · 2023-08-10

**Soundness:** 3

**Excitement:**

4: Strong: This paper deepens the understanding of some phenomenon or lowers the barriers to an existing research direction.

**Paper Topic And Main Contributions:**

This paper proposes to use BPE as a second tokenisation step in the preprocessing pipeline of symbolic music models. The authors demonstrate better performance of models trained on data that was preprocessed with this additional step (on generation tasks) and link performance gains to improved token representations (compared to token representations learned without BPE). Furthermore, their method leads to faster inference. They also discuss symbolic music classification for which their method performs comparable (but slightly worse).

**Questions For The Authors:**

A: What exactly would be problematic if the embedding dimension is larger than the vocabulary size?

**Reasons To Accept:**

Their research idea is innovative and impactful -- combining pre-tokenised inputs with BPE leads to large improvements for symbolic music generation models according to their human evaluation. Also, it positively impacts inference speed. Thus, it is worth sharing in the NLP community.

**Reasons To Reject:**

While the logical structure of the paper is sound, the writing is often sloppy and should be made more clear to boost the perceived trustworthiness of the paper:
- Often, The Transformer is mentioned in a way that suggests to the reader that it would be a specific model even though it is a model architecture (e.g. in the first sentence of the abstract and the introduction)
- The paper sometimes uses "assumption" where it should be "hypothesis" (e.g. line 505/506).
- While the examples are very helpful in Section 3 to understand BPE, not all symbols used in Algorithm 1 are explained, which might confuse a reader who doesn't know the algorithm yet. The section before the algorithm doesn't explicitly state what the base vocabulary consists of, which might again confuse an unfamiliar reader.

A challenge of the paper is that the significance of the results is not discussed. At least for the classification models and the embeddings' analysis, it would be worthwhile to evaluate models across different random seeds to get an idea of the variation of the results.

The paper sometimes makes claims that would need more explanations in my opinion:
- There is no explicit discussion why the Octuple model outperforms the BPE models on music classification
- It is mentioned multiple times in the paper that it would be problematic if the embedding dimension was larger than the vocabulary size without giving an explicit reason nor citing a source.

Some minor comments:
- I'd personally reorganise the discussion of the speed advantages of integrating BPE into the music generation pipeline. Intuitively (and as shown by the authors) tokenisation and detokenisation is slower (Table 1) while inference speed increases (Table 3). As the inference speed dominates the speed of an end-to-end system, I suspect that the method provides substantial end-to-end speed-ups, but unless I misunderstand the authors' analyses, speed is never benchmarked in an end-to-end fashion. I think such an analysis would make the paper even more convincing.

**Reproducibility:**

4: Could mostly reproduce the results, but there may be some variation because of sample variance or minor variations in their interpretation of the protocol or method.

**Reviewer Confidence:**

2: Willing to defend my evaluation, but it is fairly likely that I missed some details, didn't understand some central points, or can't be sure about the novelty of the work.

**Typos Grammar Style And Presentation Improvements:**

line 60: does -> do
line 195: Unfortunately -> Unfortunately
line 585: I'd reference directly to Table 5 instead of to Section 8.

---

> ### Author Rebuttal · Authors · 2023-08-23
>
> ### Reasons to reject
>
> Thank you for your suggestions, which are now applied to the paper.
>
> On the significance of the classification and embedding results: indeed, as we were short on space, we resorted to write concisely and may have left some ambiguity.
> Our assumption for why Octuple outperforms other baselines is that: 1) We are using a bi-directional model (no attention mask) for which the modeling task is more easily performed than with generative models with causal attention masks; 2) The sequence length is the most reduced with Octuple, which is suitable and takes advantage of the embedding space; 3) The finetuned model is trained on a classification task with a single set of labels, unlike the generation which implies several output modules / distributions / losses for octuple which are computed unconditionally from each other and adds a lot of instability. This last point explains most the differences of results for Octuple between generation and classification.
>
> On the vocabulary size / embedding size: see response below.
>
> On the end-to-end generation speed: indeed, the BPE brings a significant end-to-end generation speedup. Here is a benchmark we ran, the times are for loading MIDI --> tokenization --> generation of 512 tokens --> detokenization, with several batch sizes on a V100 GPU.
> We observe the same speed gains, with all batch sizes. With higher batch sizes, the generation is faster as the GPU takes advantage of parallelization, and case easily handle more samples for the same execution time.
> We also must note that for a given batch, the tokenization and detokenizations are performed sequentially. We could further reduce the duration by using multithreading, and encoding / decoding BPE in batch.
> We will include these results in the appendix.
>
> |Tokenization |Nb beats - 1 - TSD|Nb beats - 1 - REMI|Nb beats - 8 - TSD|Nb beats - 8 - REMI|Nb beats - 64 - TSD|Nb beats - 64 - REMI|Nb base tokens - 1 - TSD|Nb base tokens - 1 - REMI|Nb base tokens - 8 - TSD|Nb base tokens - 8 - REMI|Nb base tokens - 64 - TSD|Nb base tokens - 64 - REMI|Nb notes - 1 - TSD|Nb notes - 1 - REMI|Nb notes - 8 - TSD|Nb notes - 8 - REMI|Nb notes - 64 - TSD|Nb notes - 64 - REMI|
> |-------|------------------|-------------------|------------------|-------------------|-------------------|--------------------|------------------------|-------------------------|------------------------|-------------------------|-------------------------|--------------------------|------------------|-------------------|------------------|-------------------|-------------------|--------------------|
> |No BPE |14.0              |22.1               |53.4              |95.9               |130.6              |238.0               |95.0                    |98.8                     |410.4                   |415.3                    |1011.1                   |1046.2                    |24.9              |24.6               |106.0             |103.6              |262.6              |261.0               |
> |BPE 1k |18.9              |46.1               |108.1             |179.6              |274.1              |451.0               |176.4                   |146.8                    |734.3                   |647.1                    |1771.5                   |1591.8                    |47.7              |36.6               |195.0             |161.8              |469.0              |395.6               |
> |BPE 5k |23.4              |86.2               |97.1              |285.2              |257.3              |706.2               |221.3                   |196.7                    |950.6                   |793.7                    |2307.2                   |2020.8                    |59.4              |46.8               |255.9             |195.2              |620.3              |499.9               |
> |BPE 10k|24.8              |43.5               |117.3             |252.0              |295.1              |615.0               |243.8                   |195.0                    |955.0                   |845.5                    |2379.5                   |2115.4                    |65.8              |50.4               |258.1             |212.6              |638.0              |531.8               |
> |BPE 20k|26.3              |52.4               |125.8             |221.0              |284.6              |595.1               |269.0                   |230.2                    |1026.5                  |868.6                    |2614.4                   |2207.7                    |72.6              |58.4               |277.9             |216.6              |707.5              |547.3               |
> |PVm    |15.8              |35.4               |82.2              |119.8              |206.3              |314.1               |137.2                   |133.9                    |575.0                   |573.7                    |1470.4                   |1478.9                    |36.8              |33.3               |153.9             |148.6              |392.4              |380.9               |
> |PVDm   |23.0              |43.8               |104.5             |181.4              |305.4              |488.5               |220.9                   |209.9                    |883.7                   |866.4                    |2350.2                   |2330.3                    |61.4              |55.4               |241.3             |230.4              |642.0              |618.3               |
> |CPWord |                  |28.9               |                  |136.8              |                   |338.8               |                        |191.8                    |                        |784.5                    |                         |1922.0                    |                  |49.3               |                  |198.6              |                   |484.9               |
> |Octuple|                  |23.3               |                  |97.1               |                   |250.8               |                        |95.5                     |                        |400.1                    |                         |                          |                  |95.5               |                  |400.1              |                   |1027.5              |
>
>
> ### Questions
>
> **A:** an embedding size larger than the size of the vocabulary is not problematic in the sense that it will not degrade the performances. However it is suboptimal, or oversized as you are making the model learn to find the best localization of $V$ tokens, onto $E$ dimensions, with $E > V$.
> An example: let's say we have a set of 10k sentences, all of them with different meanings and semantic information, and we want to automatically place them in a continuous space $\mathbb{R}^E$, for instance to gather those with similarities or to measure their distances. In this case, we would want the $E$ dimensions of this space to represent some feature learned from the sentences. When choosing $E$, it makes sense to choose a value < 10k, as the 10k sentences share common features around which we want to localize them. If we set a higher $E$ value, a model can still learn and perform well, but a large number of dimensions will simply be unused.
> In our case, models trained with small vocabularies (< to embedding size) still perform well, but the model could have learned a much larger number of token / embeddings, while performing as well or better as reported by our results in tables 2 to 5. Table 3 shows that the generative models still use (i.e. outputs high probabilities during inference) most of the tokens (99% with 30k tokens) when the vocabulary grows.
>
> We thank you for these constructive questions / remarks that will undoubtedly strengthen the quality / affirmations of the paper.

---

### Official Review · Reviewer_SeWr · 2023-08-11

**Soundness:** 4

**Excitement:**

4: Strong: This paper deepens the understanding of some phenomenon or lowers the barriers to an existing research direction.

**Paper Topic And Main Contributions:**

This paper studies the representation of symbolic music in generation and classification tasks. The author proposes to utilize Byte Pair Encoding (BPE) to reduce the sequence length for generation and classification. The paper compares the proposed method with several dimension reduction baselines as well as the tokens without BPE. Finally, the limitation of using BPE is discussed in the paper.

**Reasons To Accept:**

- The average token number can be reduced by 50% by utilizing the BPE on top of the normal tokenizer while achieving better performance. This feature can make future studies on music generation and classification less computationally expensive, and the representation can be easier to modelling on.
- The proposed method can expand the vocabulary set and make better use of the embedding dimension of the language model.

**Reasons To Reject:**

- Despite the experiment being interesting and showing promising result, the research mostly build upon the well-studied modules and appears to be somehow experimental.

**Reproducibility:**

4: Could mostly reproduce the results, but there may be some variation because of sample variance or minor variations in their interpretation of the protocol or method.

**Reviewer Confidence:**

3: Pretty sure, but there's a chance I missed something. Although I have a good feel for this area in general, I did not carefully check the paper's details, e.g., the math, experimental design, or novelty.

---

> ### Author Rebuttal · Authors · 2023-08-23
>
> We would be happy to receive explanations about how the research "appears to be somehow experimental", and why is it a reason to reject it. We assume that this point motivated your soundness score. Having more details would help up to understand better your decision, and how we can improve the paper.

---

### Meta-Review · Area_Chair_GfNc · 2023-09-17

**Recommendation:** 4

**Metareview:**

This paper introduces an innovative application of Byte Pair Encoding (BPE) in the realm of music scenarios. The proposed algorithm significantly enhances inference speed and result quality, only with a short preprocessing step. This research holds valuable insights for researchers and engineers engaged in symbolic music.

The reviewers have agreed on the innovation and impact of this research. By reducing the average token count by 50% while achieving improved performance and faster inference speed, the proposed method has the potential to reduce computational overhead in future studies on music generation and classification, making representation modeling more accessible. Furthermore, the paper is well-written, providing a comprehensive package with code, audio samples, and results on a dedicated website.

---

### Decision · Program_Chairs · 2023-10-07

**Decision:**

Accept-Main

**Comment:**

This paper introduces an innovative application of Byte Pair Encoding (BPE) in the realm of music scenarios. The proposed algorithm significantly enhances inference speed and result quality, only with a short preprocessing step. This research holds valuable insights for researchers and engineers engaged in symbolic music.

The reviewers have agreed on the innovation and impact of this research. By reducing the average token count by 50% while achieving improved performance and faster inference speed, the proposed method has the potential to reduce computational overhead in future studies on music generation and classification, making representation modeling more accessible. Furthermore, the paper is well-written, providing a comprehensive package with code, audio samples, and results on a dedicated website.